# The Association between Nutrition, Physical Activity, and Cardiometabolic Health at 6 Months following a Hypertensive Pregnancy: A BP^2^ Sub-Study

**DOI:** 10.3390/nu15153294

**Published:** 2023-07-25

**Authors:** Camilla Hirsch, Lynne Roberts, Justine Salisbury, Elizabeth Denney-Wilson, Amanda Henry, Megan Gow

**Affiliations:** 1Discipline of Women’s Health, School of Clinical Medicine, UNSW Medicine and Health, University of New South Wales, Sydney, NSW 2052, Australia; c.hirsch@student.unsw.edu.au (C.H.); amanda.henry@unsw.edu.au (A.H.); 2Women’s and Children’s Health, St. George Hospital, Sydney, NSW 2217, Australia; lynne.roberts2@health.nsw.gov.au; 3St. George and Sutherland Clinical Campus, School of Clinical Medicine, UNSW Medicine and Health, University of New South Wales, Sydney, NSW 2052, Australia; 4NSW Ministry of Health, Sydney, NSW 2060, Australia; justine.salisbury@health.nsw.gov.au; 5Susan Wakil School of Nursing, University of Sydney, Camperdown, Sydney, NSW 2006, Australia; elizabeth.denney-wilson@sydney.edu.au; 6Discipline of Paediatrics, School of Clinical Medicine, UNSW Medicine and Health, University of New South Wales, Sydney, NSW 2052, Australia; 7Children’s Hospital Westmead Clinical School, University of Sydney, Sydney, NSW 2006, Australia

**Keywords:** blood pressure, body mass index, hypertensive disorders of pregnancy, lifestyle, nutrition, physical activity, postpartum, waist circumference

## Abstract

Hypertensive disorders of pregnancy (HDP) complicate 5–10% of pregnancies, with resultant lifelong increased risks of cardiovascular disease (CVD). We aimed to describe lifestyle behaviours at 6 months post-HDP in four HDP subgroups, and their association with markers of cardiometabolic health. Subgroups were chronic hypertension (CH), gestational hypertension (GH), preeclampsia, and preeclampsia superimposed on chronic hypertension (CH + PE). The BP^2^ study is a multi-site, three-arm, randomised controlled trial. At 6 months postpartum, the NSW Population Health Survey and BP^2^ surveys collected lifestyle behaviours and demographic data. Body mass index (BMI), waist circumference, and blood pressure (BP) were also assessed. Descriptive statistics, ANOVA and Spearman’s correlation coefficients were used. Of 484 women (16% CH, 23% GH, 55% preeclampsia, and 6% CH + PE), 62% were overweight or obese. Only 6% met the recommended five vegetable and two fruit serves per day, and 43% did not meet the recommended 150 min of moderate–vigorous physical activity in five sessions per week. Adherence to both diet and physical activity recommendations was correlated with more favourable cardiometabolic outcomes, including lower BMI, waist circumference, and systolic and diastolic BP. Lifestyle interventions that improve diet and physical activity post-HDP are needed to reduce BP, BMI, and long-term CVD in this high-risk population.

## 1. Introduction

Hypertensive disorders of pregnancy (HDP) complicate 5–10% of pregnancies worldwide [1,2,3,4], including 3% gestational hypertension (GH), 0.2–5% chronic hypertension (CH) [5], and 2–5% preeclampsia [3,6], including some with preeclampsia superimposed on chronic hypertension (PE + CH). HDP are a leading cause of female morbidity and mortality worldwide [7], with preeclampsia causing >70,000 maternal and >500,000 perinatal deaths annually [1]. They are also a leading cause of preterm and small for gestational age births [1,8]. They are also a leading cause of preterm and small for gestational age births [1,8]. The main risk factors for HDP include obesity, prior HDP, excessive gestational weight gain, and family history of hypertension [1,4,9].

Following HDP, women are at an increased risk of several long-term chronic conditions which develop at higher rates and years earlier than matched controls [8,10]. The most significant and well-established is the 2–3 times increased cardiovascular disease (CVD) risk [2,4,6], including coronary artery disease, heart failure, peripheral vascular disease, dementia, stroke, and early cardiovascular-related death [8,9,10], even after adjusting for factors such as education, marital status, smoking, and obesity [8,11]. HDP is associated with a 12–25× increased risk of chronic hypertension within the first year postpartum, 10× increased risk in the first 10 years, and 3–4× increased risk ongoing [6,12]. Furthermore, a woman’s relative risk of type II diabetes is more than doubled, and risk of chronic renal disease and dyslipidaemia increases several-fold [2,4,13]. This risk is of critical importance as CVD is the leading cause of death for women [14], responsible for 16% and 25% of maternal deaths in high and low–middle income countries, respectively [7].

As suboptimal lifestyle behaviours of low fruit and vegetable intake, low levels of physical activity, and weight change are some of the key risk factors for maternal CVD [15], HDP allows early identification of high-risk women and potential to modify lifelong lifestyle practices to reduce adverse cardiometabolic outcomes [10,16,17]. The International Society for the Study of Hypertension in Pregnancy (ISSHP) recommend that post-HDP, women should receive counselling about their increased risk of future HDP and CVD, as well as the benefits of adopting a healthy lifestyle including achieving and maintaining ideal weight [6,10,16]; however, this is not currently built into standard postpartum management globally [18,19]. Diets such as Dietary Approaches to Stop Hypertension and exercise recommendations (150 min per week of moderate-to-vigorous activity in five sessions) [16,20] are cost-effective and reputable CVD prevention strategies in the general population [2], and thus should be encouraged for women following a pregnancy affected by HDP [4].

Qualitative research following pregnancy [21] and preeclampsia [22] highlights that many women postpartum experience suboptimal diet and physical activity rates compared to pre-pregnancy [20,21], despite increased motivation during this period [22,23]. Furthermore, women post-HDP were found to have more micronutrient intake insufficiency compared to normotensive controls [4], where reduced intake of specific micronutrients have been associated with increased CVD risk [24,25].

Lifestyle interventions in women following a pregnancy affected by preeclampsia were found to effectively increase knowledge of future CVD risk [22] and engagement with physical activity [26], whilst decreasing fat intake [17], body mass index (BMI), and waist-to-hip ratio [17]. Future studies are required to better understand lifestyle behaviours following HDP and their relation to weight change and other CVD markers in order to guide lifestyle intervention development in this population [10,17,27,28].

### Aims

Describe lifestyle behaviours (nutrition and physical activity) of women 6 months post-HDP.Compare lifestyle behaviours between four HDP subgroups (CH, GH, preeclampsia, and CH + PE).Identify associations between lifestyle behaviours and cardiometabolic outcomes including body mass index (BMI), waist circumference, and blood pressure (BP).

## 2. Materials and Methods

This project is a sub-study of the prospective Blood Pressure Postpartum (BP^2^) study, a three-arm randomised controlled trial (RCT) being conducted at six hospitals in metropolitan Sydney, Australia [6]. These hospitals service a socio-demographically diverse population, with around 20,000 births each year [6]. The BP^2^ study aims to evaluate the effectiveness of three postpartum lifestyle intervention and follow-up strategies (Appendix A), in improving women’s health following HDP. Group one: follow-up with primary healthcare provider, group two: follow-up at dedicated postpartum clinic including consultation with physician and dietitian, and group three: as for group two plus automatic enrolment in 6-month lifestyle coaching through the Get Healthy Service (GHS) NSW Health. The GHS is a free, phone-based lifestyle behaviour change program delivered by trained dietitians and exercise physiologists, wherein participants receive coaching/assistance to achieve their lifestyle goals with up to 10 phone calls over a six-month period.

The BP^2^ study’s primary outcomes are maternal BP, weight, and waist circumference. Secondary outcomes relevant to this sub-study include maternal lifestyle behaviours, health motivation, and fat mass percentage. For full methodology, please refer to the BP^2^ study protocol (2020) [6].

### 2.1. Ethics Approval

The BP^2^ study has been approved by the South-Eastern Sydney Local Health District (Ref: 18/193, REGIS: 2019/ETH04732) and was prospectively registered in the Australian and New Zealand Clinical Trials Registry (ACTRN12618002004246).

### 2.2. Participant Eligibility and Recruitment

Women diagnosed with HDP, ≥18 years of age, who gave birth at study hospital sites were eligible to participate in the study. Women whose pregnancy resulted in still-birth or neonatal death were ineligible, as were women who were unavailable for follow-up, suffered from a severe mental health condition, or developmental disability which precluded informed consent. Participation of women from non-English speaking backgrounds was encouraged through interpreter services and translated study materials [6]. Women were approached by site-coordinators prior to postpartum hospital discharge and/or 4–5 months postpartum, with eligible and willing participants giving written informed consent.

BMI was calculated using the standardised formula [29], with overweight defined as a BMI of ≥25 kg/m^2^ and <30 kg/m^2^, and obesity defined as a BMI of ≥30 kg/m^2^ [30].

Hypertension in pregnancy was defined as an average systolic BP of ≥140 millimetres of mercury (mmHg) and/or a diastolic BP of ≥90 mmHg, over at least two measurements [10]. CH is hypertension diagnosed before pregnancy or prior to 20 weeks’ gestation whilst GH is hypertension arising after 20 weeks’ gestation, without other findings suggestive of preeclampsia. Preeclampsia is hypertension arising after 20 weeks’ gestation with the presence of any of the following conditions: proteinuria, maternal end-organ dysfunction (neurological, pulmonary, haematological, renal, or hepatic), and/or uteroplacental dysfunction (fetal growth restriction or intrauterine fetal death). Preeclampsia may be superimposed on CH i.e., CH + PE [10].

### 2.3. Sub-Study Data Collection

Data collected from medical records by site researchers included parity, HDP diagnosis, pregnancy, and birth details. Maternity databases were accessed by the research midwives at each study site to extract the required data. Lifestyle behaviours were assessed using the NSW Population Health Survey, which was developed by the NSW government to assess health behaviours over the phone (Appendix A) [31]. In this survey, women self-reported their diet (fruit, vegetable, salty snack, fried potato, processed meat, take away, alcohol, water, soft drink, and fruit juice intake per day/week/month, as well as type of milk consumed), physical activity behaviours (all moderate to vigorous physical activity, walking, vigorous chores/yard work, and strength training each day/week/month) and smoking status [6].

The BP^2^ study survey (Appendix A), a self-reported online survey, collected demographic data including age, ethnicity, education, BMI at the first antenatal appointment for the index pregnancy, medications, and breastfeeding status. The researcher went through the free text field for ‘Current medications’ and tabulated participant responses as ‘Yes’ or ‘No’ for current antihypertensive use in a new data column for analysis. This survey also included the validated Adherence to a Healthy Lifestyle Questionnaire [32] which assessed motivation, perceived barriers, and satisfaction regarding lifestyle behaviours.

Following completion of the pre-randomisation questionnaires (BP^2^ study survey and NSW Population Health Survey), participants were randomised at approximately 6 months postpartum into one of the study groups. Maternal cardiovascular risk was assessed at a 6-month visit by site-coordinators and medical students, or by the woman’s General Practitioner (GP), using measurements of maternal weight, body composition, BMI, waist circumference and BP, all taken using standard procedures [6].

### 2.4. Outcome Measures

This sub-study’s primary outcomes were diet and physical activity behaviours 6-months following HDP [6]. The covariates investigated for their association with primary outcomes included maternal BMI, fat mass percentage, waist circumference, and BP.

### 2.5. Statistical Analysis

Data from all study sites were exported from the central REDCap BP^2^ database into Microsoft Office Excel version 16.64. The central BP^2^ database on REDCap is a centralised location for all collected data that research staff from each study site input data into. Regular data audits were performed and then study data cleaned/missing data sought prior to analysis by the BP^2^ study manager. Data cleaning and organisation allowed identification of errors and gaps, which were investigated and corrected prior to analysis.

Statistical analysis was performed using IBM SPSS Statistics version 27.0 (Chicago, IL, USA). Descriptive statistics were performed using mean ± standard deviation for normally distributed continuous data, and median with interquartile range for not normally distributed data. Distribution was determined using a combination of Shapiro–Wilk test for the null hypothesis, skewness and kurtosis z-values, and visual inspection of frequency histograms and boxplots. Number and percentages were used for categorical data.

ANOVA tests were used to compare outcomes overall and between HDP subgroups. Correlations between lifestyle behaviours and cardiometabolic outcomes were analysed using Spearman’s correlation coefficients, as lifestyle outcome data were not normally distributed. Categorical outcomes were recoded numerically as binary (full fat milk yes/no and currently breastfeeding yes/no) or in increasing order (smoking status non/past/current and electronic cigarette status non/past/current) to allow for evaluation of these correlations. The level of significance in this exploratory study is a *p*-value < 0.05.

## 3. Results

In total, 2652 women diagnosed with HDP were screened and provided Participant Information Statement and Consent prior to postpartum hospital discharge. Of these, 342 were excluded as they did not meet the inclusion criteria and 2310 were contacted 5 months postpartum to either confirm prior consent, provide further trial information, or gain informed consent. Of these, 915 declined to participate or declined further information, and 840 were not contactable. In total, 555 women consented to participate. All participants were recruited and randomised for the BP^2^ study between 30 January 2019 and 27 July 2022. Those 484 women with completed 6-month data were included in the present analysis (Figure 1).

### 3.1. Study Population Characteristics

Of the 484 women, 76 (16%) had CH, 113 (23%) had GH, 268 (55%) had preeclampsia, and 27 (6%) had CH + PE in the index pregnancy. Table 1 describes participant characteristics and differences between HDP subgroups.

Maternal age averaged 34.5 ± 5.2 years, the majority of women were Caucasian (59%), and for over two thirds, the index pregnancy was their first birth experience (67%). Women following a pregnancy affected by preeclampsia were significantly younger (mean age of 33.5) and had higher rates of first pregnancy (77%) when compared to other subgroups. There were no differences in ethnicity across HDP subgroups. Overall, demographics were similar between subgroups; however, differences were seen between highest level of education, gestation at diagnosis and birth, BMI at first antenatal appointment, maternal days spent in hospital at birth, and number of children at time of follow-up. At the first antenatal appointment for the index pregnancy, 50% of participants fell into the overweight or obese BMI range, with the highest rates for both in the CH subgroup. Those with CH + PE were more likely to have completed a university degree (*p* = 0.002), and those with preeclampsia or CH + PE gave birth earlier (*p* < 0.001) and spent longer in hospital during admission for the birth of their index pregnancy (*p* < 0.001). GH was diagnosed at later gestations than other subtypes, and the subtypes with a higher number of children at time of follow-up were CH and CH + PE. At 6 months postpartum, 8 women (2%) were current daily smokers, and 107 were past smokers (22%). A total of 297 women were breastfeeding (62%), and 141 women (29%) had stopped since birth at a median of 3 months postpartum. No significant differences were found across HDP subgroups for these outcomes.

### 3.2. Primary Lifestyle Outcomes

Table 2 describes the lifestyle outcomes of women post HDP at 6 months postpartum, as collected by the NSW Population Health Survey. In the previous 4 weeks, 398 women (82%) reported their physical health was good or better, whilst 84 (17%) reported that their health was fair or worse (with two “don’t know”).

#### 3.2.1. Nutrition

Median daily vegetable intake was two serves (IQR 2.0) with only 9% (*n* = 44) meeting the recommended five serves per day. Fruit was consumed at a daily median of one serve (IQR 1.0) with 46% meeting the recommended two fruit serves per day (*n* = 220). Only 6% of women were meeting both the recommended five vegetables and two fruits per day (*n* = 31). Women in the CH and CH + PE categories had significantly lower rates of meeting recommended fruit intake, and increased sports/soft drink intake compared to other subgroups (*p* < 0.05). ‘Amount of sugar’ was found to be most influential in food choices for the majority of women (*n* = 258, 53%), with ‘carbohydrates’ and ‘fat’ the next most influential. In assessing nutritional understanding, 48% of women reported that they did not know how many kilojoules (kJ) adults needed per day on average (*n* = 232), and only 26% identified the correct 8700 kJ/day.

The median number of standard alcoholic drinks consumed per week was zero (interquartile range (IQR) 1.5), and 55 women (20%) had consumed more than four alcoholic drinks on one occasion at least once in the preceding four weeks.

#### 3.2.2. Physical Activity

The median total time spent doing moderate–vigorous physical activity was 210 min per week (IQR 270 min), including both moderate–vigorous exercise and continuous walking. Thus, 64% of women were found to be meeting the recommended 150 min per week of physical activity (*n* = 372) in a median of six sessions each week (IQR 6.0), with 57% meeting recommendations for both 150 min per week in at least five sessions. The lowest rates of physical activity were participants in the CH and CH + PE subgroups, compared with those of the GH subgroup who had the highest rates of weekly activity. Only 5% of women were found to be meeting both recommendations for physical activity as well as recommendations for daily fruit and vegetable intake (*n* = 22).

Overall, nutrition and physical activity outcomes were similar across HDP subgroups, except for time spent doing vigorous chores and moderate–vigorous physical activity; however, these differences should be interpreted with caution considering medians were zero across most subgroups for these outcomes.

### 3.3. Weight Outcomes

Weight outcomes at 6 months postpartum are outlined in Table 3. Median BMI fell within the overweight range, with 62% of participants falling into the ranges consistent with overweight or obesity, and over 50% of women in the CH subgroup falling into the range of obesity. The preeclampsia subgroup was found to be the closest to a healthy median BMI (*p* < 0.001). At follow-up, 29% of women had returned to their self-reported pre-pregnancy BMI or lower, with no significant difference across HDP subgroups.

Median waist circumference and fat mass percentage were also highest in the CH subgroup (*p* < 0.001), whilst lowest median waist circumference was in the preeclampsia subgroup. Median BPs at 6 months postpartum were 122 mmHg systolic (IQR 15 mmHg) and 81 mmHg diastolic (IQR 14 mmHg), with over half the participants reading above the recommended cut-offs of 120 mmHg systolic and/or 80 mmHg diastolic. Median BPs in the CH and CH + PE subgroups were significantly higher than the other groups (*p* < 0.001), despite 65% and 85% of participants taking antihypertensives in these groups, respectively.

### 3.4. Correlations between Lifestyle Behaviours and Cardiometabolic Outcomes

Pearson’s correlation coefficients are included in Appendix A.

Greater daily vegetable serves were associated with reduced fat mass percentage, and meeting the daily vegetable intake recommendation was associated with a significant reduction in waist circumference. Meeting the recommended daily fruit intake was associated with a reduction in BMI, waist circumference, and average diastolic BP. Diets with greater intake of processed meats, hot chips, salty snacks, soft drinks, and takeaway occurrences were each independently associated with increased BMI and waist circumference. Greater number of takeaway occurrences and soft drink intake were further correlated with increased fat mass percentage. Increased salty snack serves was associated with increased systolic and diastolic BP, whilst higher soft drink intake correlated with increased systolic BP only.

Meeting the recommended weekly 150 min of physical activity and five sessions per week both independently correlated with lower BMI, waist circumference, fat mass percentage, and diastolic BP. Stronger correlations with these cardiometabolic outcomes were found for women fulfilling both physical activity recommendations concurrently. Meeting both physical activity (150 min per week in at least five sessions) and dietary recommendations (five vegetables and two fruits per day) was associated with reduced BMI, waist circumference and systolic and diastolic BP.

Outcomes of greater walking and vigorous physical activity were independently associated with improved outcomes such as reduced BMI and waist circumference. However, greater minutes of vigorous household chores and vigorous yard work were associated with worsened cardiometabolic outcomes, such as increased waist circumference, fat mass percentage, and systolic BP.

More alcohol intake occurrences were associated with reduced BMI and waist circumference, whereas greater number of standard drinks consumed each week was only associated with reduced BMI. Alcohol intake was found to be greater in women with fewer children.

Smoking correlated with increased BMI, waist circumference, and fat mass percentage, whilst breastfeeding was found to be weight protective in all domains and cardioprotective with added reduction in systolic BP. Having a greater number of children was associated with generally reduced adherence to recommended weekly physical activity and increased weekly household chores and yard work.

These correlations were statistically significant.

## 4. Discussion

This sub-study has revealed the suboptimal diet and physical activity of women 6 months following HDP, with correlations clearly present between healthy lifestyle behaviours, and improved weight and cardiometabolic health. These findings emphasise the importance of lifestyle interventions and education in this population to manage risk factors and prevent long-term cardiometabolic sequalae in an already vulnerable group.

### 4.1. Nutrition

In line with our findings that under 10% of women were meeting the recommended five daily vegetable serves, less than half met the daily two fruit serves recommendation and 6% met both; limited studies previously assessing dietary intake in new mothers have similarly shown diet quality to be inadequate postpartum [4,33,34,35]. A study by van der Pligt et al. [33] similarly found only 9% of 448 women postpartum met the recommendation of five vegetable serves per day whilst 55% met the two fruits daily recommendation [33]. This is concerning from a public health perspective given the well-documented benefits of healthy lifestyle behaviours on reducing chronic disease risk [33,36], which is particularly important for women following a pregnancy affected by HDP given their increased long-term risk. These findings reveal the importance of education and developing intervention strategies to promote adequate fruit and vegetable intake in the general postpartum population, with special emphasis in the post-HDP population [33], where healthy diet practices may be even more critical.

### 4.2. Physical Activity

Compared to our median 210 min per week of moderate–vigorous physical activity and 180 min per week of walking, with 57% of women achieving this in at least five sessions, van der Pligt’s analysis within the general postpartum population found that moderate–vigorous physical activity and walking averaged much higher (351 min per week and 252 min per week, respectively), with 63% meeting recommendations in at least five sessions [33]. These findings suggest that the post-HDP population may be less physically active than their counterparts in the general postpartum population, which is concerning as physical activity is a key component for CVD risk reduction. In line with previous studies conducted in new mothers [37], we found that walking was the greatest contributor to weekly physical activity which should be encouraged for all women following a pregnancy affected by HDP, as it provides an accessible, low-cost, and low-risk activity which significantly lowers BMI and chronic disease risk [33,38]. Insufficient physical activity in over 40% of our participants may be attributed to lack of motivation, time, and energy whilst caring for an infant [39,40,41], consistent with existing literature outlining the decline in activity in the postpartum period [20,21]. This theory is consistent with our findings that mothers with more children had an associated reduction in time spent exercising, walking, and being physically active, with increased time spent doing household chores and yard work. An intervention tailored to the time restraints of new mothers may significantly improve physical activity postpartum and lifelong, to provide lasting reductions in CVD risk in this more vulnerable post-HDP population.

### 4.3. Weight Outcomes

Median BMI at 6 months following HDP was in the overweight range, with 30% of women having obesity [30]; however, as 16% of our sample were of Asian ethnicity, caution should be taken in interpreting these classifications which are defined for the general population. Different BMI cut-offs may have been more appropriate for these women due to variations in muscle and fat mass [42]. Furthermore, at the time of follow-up, less than 30% of participants had returned to their self-reported pre-pregnancy BMI; thus, rates may appear inflated due to maintained gestational weight gain. However, this too should be interpreted with caution due to reliance on participants’ accurate recall. Median BMI, waist circumference, and fat mass percentage were all highest in the CH subgroup in our analysis, which may be attributed to the shared risk factors between these outcomes and CH, or that poor cardiometabolic profiles contribute to the pathophysiology of their underlying condition. Other studies which describe the weight profiles of women specifically following HDP are scarce. McLennan et al. [43] found that 6 months following preeclampsia, participants had higher BMI and fat mass percentage compared to normotensive controls. These women had mean BMI in the overweight range [43] and had a fat mass percentage of 40.7 ± 7.4 [43], which was similar but slightly higher than our population’s 37.0 ± 8.4. In comparison to Bijlholt et al.’s study [44] amongst a non-specific cohort of women 6 weeks postpartum [44], our participants following HDP were of higher adiposity than their general postpartum population, even though controlling adiposity is critical post-HDP to minimise long-term CVD risk. Median systolic and diastolic BPs were approaching the hypertensive range [10], despite many women taking antihypertensives. These findings are concerning as managing BMI, body composition, and BP is critical in controlling CH and CVD risk. Interventions which target risk education and weight management postpartum may prove invaluable in minimising future cardiometabolic sequalae [27,45,46,47,48,49] including subsequent HDP, as Tano et al. demonstrated that interpregnancy BMI gain and pre-pregnancy BMI have significant associations with HDP in subsequent pregnancies [50].

### 4.4. Correlations between Lifestyle Behaviours and Weight Outcomes

#### 4.4.1. Nutrition

Meeting recommended fruit and vegetable intake were independently associated with a range of improvements in cardiometabolic health outcomes, such as BMI, waist circumference, fat mass percentage, and diastolic BP, whilst rates of suboptimal dietary behaviours were associated with worse health outcomes. Significant correlations between daily fruit and vegetable serves with fat mass percentage, waist circumference, and BMI were to be expected as they may indicate women eating relatively healthier diets which have been previously reported in the general population to have well-established positive impacts on adiposity and BMI [2,16,34,51]. Interestingly, vegetable serves per day was not individually associated with significant reductions in BP or BMI, which may be due to a blunting effect as so few of our participants were meeting recommended vegetable intake. The impact of daily fruit serves on diastolic BP can be potentially attributed to the anti-oxidative properties of fruits, which have been shown to significantly reduce BP [52]. Future interventions which target fruit and vegetable intake post-HDP have the potential to optimise cardiometabolic health and mitigate increased CVD risk in this population [16,33]. As expected, increased rates of suboptimal diet behaviours were all found to have significant associations with increased BMI and waist circumference, consistent with findings in the general population that foods high in salt and sugar are associated with raised BMI, hypertension, and overall worsened cardiometabolic outcomes [53,54,55]. The correlation between increased salty snack serves and raised systolic and diastolic BP was also expected, as sodium intake has been shown to have a significant association with increased BP [4,53,55]. In women following a pregnancy affected by HDP, who are already at an increased risk of worsened cardiometabolic risk profiles, controlling dietary sodium and processed sugar intake via education or intervention could significantly improve their cardiometabolic health, and potentially reduce long-term CVD risk [53,54,55].

#### 4.4.2. Physical Activity

Physical activity was found to be significantly correlated with improvements in BMI, waist circumference, fat mass percentage, and diastolic BP, consistent with the well-documented benefits of physical activity on cardiovascular risk profiles in both general and non-specific postpartum populations [20,38,56,57,58]. Independent correlations between increased time spent doing vigorous household chores and yard work with increases in waist circumference, fat mass percentage, and systolic BP were not expected; however, this may demonstrate that those women gaining their ‘minutes’ of physical activity doing household chores may not be reaping the same benefits as those women gaining their ‘minutes’ doing other types of more purposeful physical activity. We excluded these outcomes from calculations for moderate–vigorous physical activity due to inconsistencies and potential subjectivity in reporting which would risk overestimation. Thus, this finding indicates the potential benefit of promoting purposeful physical activity to create the most significant impact on cardiometabolic profile post-HDP. The only lifestyle outcome which correlated with reduced systolic and diastolic BP was adherence to both dietary and physical activity recommendations, despite only 5% of women achieving this. This is not unexpected and is likely due to the compounding individual benefits of each lifestyle factor in cardiometabolic health.

Healthy lifestyle interventions targeting improved diet, physical activity, smoking cessation, and reduced alcohol intake will contribute to reduced long-term cardiovascular risk.

### 4.5. Limitations

One key limitation is the cross-sectional design of this sub-study, which did not allow for longitudinal analysis of outcome measures over time. Ongoing follow-up as part of the broader BP^2^ cohort study is needed to examine the sustainability of these lifestyle behaviours and their impact on cardiometabolic risk over time [4,23,28], to further guide the development of evidence-based guidelines for CVD risk reduction following HDP [27]. BMI classifications in the present study may not be appropriate for certain ethnicities, as international heterogeneity was not accounted for [57]; thus, future studies could delineate based on participant ethnicity to avoid misclassification. Assessing nutrition, physical activity, and pre-pregnancy BMI in self-reported questionnaires allowed for potential subjectivity in reporting; however, this facilitated our larger-scale evaluation and comparison with cardiometabolic outcomes. This study did not include a normotensive control group, which could be incorporated into future studies of its kind to further enhance the validity of these findings for women specifically following HDP. Lifestyle behaviours may have been impacted by this study’s COVID-19 context which has been associated with reduced physical activity levels [59]; thus, future studies confirming the reliability of these findings are necessary. Furthermore, due to evolving hospital restrictions and contextual apprehension to present to study sites/general practitioners during the COVID-19 pandemic, cardiometabolic outcome data were self-reported or missing for a small number of women. Finally, lifestyle questionnaires did not collect information about perceived drivers and barriers of physical activity and nutrition; thus, incorporating these into future studies could prove instrumental in informing and guiding the development of these lifestyle interventions.

### 4.6. Strengths and Future Directions

As far as we are aware, the present study is the first to describe nutritional and physical activity behaviours, cardiometabolic profiles, and the correlations between them for women early post various types of HDP. HDP diagnoses were made using the ISSHP definitions, allowing for accuracy and consistency across sites. The study’s sample size is also a strength, as well as its use of participants from six Sydney, Australia hospitals, representing an ethnically diverse population, enhancing the generalisability of study findings to the wider Australian and global postpartum population. The findings have significant clinical and public health implications, as they have the potential to guide postpartum management and development of interventions in the post-HDP population, to improve healthy lifestyle behaviours, and optimise cardiometabolic health to ultimately reduce future CVD risk.

A qualitative study exploring education preferences of Australian women regarding long-term health after a HDP revealed that education should be a key component of postpartum care, with easily accessible information through an electronic resource suggested as having the greatest potential [60]. Ryan et al. and Makama et al.’s systematic reviews, and Almli et al.’s focus groups, all identified that lacking in physical time and energy as well as prioritisation of children over themselves were key barriers to maintaining a healthy lifestyle postpartum [39,40,41]. Combining the findings within these studies and ours, accessible and convenient access to education such as through digital interventions are suggested to avoid the barriers faced in early motherhood, such as verified social media pages and the Get Healthy Service phone service, which can be explored at any time.

## 5. Conclusions

Six months following HDP, most women were not meeting recommended daily fruit and vegetable serves, and almost half were not doing the recommended amount of weekly physical activity. As expected, associations between healthy lifestyle behaviours and better weight outcomes were significant, with the converse also found to be true. As the link between HDP and future cardiovascular-related morbidity and mortality is well established, it is critical that these lifestyle behaviours are optimised to minimise future CVD risk and subsequent HDP. These findings may inform the development of lifestyle interventions and management plans for women postpartum, through identification of areas with significant effects on weight and those in need of greatest improvement. Standardised lifestyle interventions following HDP will be of vital public health importance in developing effective risk minimisation strategies for this higher-risk population; however, these are yet to be formally evaluated in both the short- and long-term.

## Figures and Tables

**Figure 1 nutrients-15-03294-f001:**
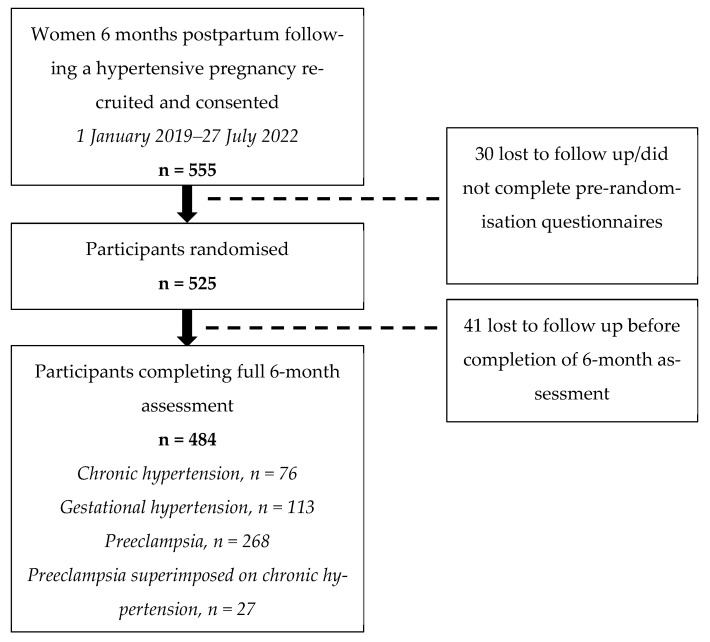
Recruitment and final number included for this sub-study’s analysis. Abbreviations: n, number.

**Table 1 nutrients-15-03294-t001:** BP^2^ study participant demographics per hypertensive subgroup, at 6 months postpartum.

Demographic (Missing)	Total (*n* = 484)	CH (*n* = 76)	GH (*n* = 113)	Preeclampsia (*n* = 268)	CH + PE (*n* = 27)	*p*-Value
	** *Number (%)* **					
***Ethnicity*** (0)						
Caucasian	285 (59)	41 (54)	79 (70)	150 (56)	15 (56)	0.06
Asian	79 (16)	18 (24)	15 (13)	39 (15)	7 (26)	0.1
Aboriginal or Torres Strait Islander	3 (1)	0 (0)	1 (1)	2 (1)	0 (0)	0.84
Polynesian	8 (2)	3 (4)	0 (0)	5 (2)	0 (0)	0.18
European	33 (7)	2 (3)	9 (8)	21 (8)	1 (4)	0.37
Middle Eastern	9 (2)	0 (0)	1 (1)	7 (3)	1 (4)	0.34
African	11 (2)	4 (5)	1 (1)	6 (2)	0 (0)	0.2
Other ^⊚^	53 (11)	8 (11)	6 (5)	36 (13)	3 (11)	0.15
***Highest education*** (4)						0.02 *
University degree	326 (68)	42 (56)	82 (74)	180 (67)	22 (82)	
Trade/certificate/diploma	108 (23)	18 (24)	25 (23)	61 (23)	4 (15)	
Secondary school	46 (10)	15 (20)	4 (4)	26 (10)	1 (4)	
***Index pregnancy first baby*** (0)	326 (67)	35 (46)	75 (66)	205 (77)	11 (41)	<0.001 *
***Index pregnancy planned*** (3)	384 (80)	60 (79)	92 (82)	212 (80)	20 (74)	0.81
***Index pregnancy twin pregnancy*** (0)	16 (3)	0 (0)	2 (2)	14 (5)	0 (0)	0.06
***Gestation at HDP diagnosis*** (1)						<0.001 *
Pre-existing	39 (8)	39 (51)	0 (0)	0 (0)	0 (0)	
During pregnancy	409 (85)	37 (49)	99 (88)	246 (92)	25 (93)	
Completed weeks—median (IQR)	36.0 (6.0)	15.0 (5.0)	37.0 (4.0)	36.0 (5.0)	35.0 (7.0)	
Intrapartum	5 (1)	0 (0)	2 (2)	3 (1)	0 (0)	
Postpartum	31 (6)	0 (0)	12 (11)	19 (7)	2 (7)	
	** *Median (IQR)* ** ** * ^⋄^ * **					
***Index pregnancy BMI at first antenatal appointment*** (kg/m^2^) (5)	25.0 (7.3)	29.0 (8.6)	25.1 (6.0)	23.9 (6.9)	25.6 (6.5)	<0.001 *
Overweight; BMI 25.0–29.9, n (%)	130 (27)	24 (32)	34 (30)	65 (24)	7 (26)	
Obese; BMI ≥ 30, n (%)	112 (23)	31 (42)	24 (21)	50 (19)	7 (26)	
***Gestation at birth (completed weeks)*** (0)	38.0 (3.0)	38.0 (1.8)	39.0 (1.0)	37.0 (4.8)	36.0 (4.0)	<0.001 *
***Maternal hospital days stay at birth*** (0)	6.0 (3.0)	5.0 (3.0)	5.0 (3.0)	7.0 (4.0)	8.0 (4.0)	<0.001 *
	** *Mean ± standard deviation* **					
***Maternal age at 6 M assessment**(years)*** (0)	34.5 *±* 5.2	36.1 ± 5.1	35.1 ± 4.8	33.5 ± 5.3	36.4 ± 4.0	<0.001 *
	** *Number (%)* **					
***Currently in a relationship*** (4)	442 (91)	70 (93)	105 (94)	242 (91)	25 (93)	0.79
***Breastfeeding at 6 months*** (3)						0.40
Yes, n (%)	297 (62)	47 (62)	72 (64)	163 (61)	15 (56)	
No (never breastfed), n (%)	43 (9)	8 (11)	5 (5)	25 (9)	5 (19)	
No (I’ve stopped now), n (%)	141 (29)	21 (28)	35 (31)	78 (29)	7 (26)	
When stopped (months after birth), median (IQR)	3.0 (2.4)	3.0 (3.6)	3.5 (2.7)	3.0 (2.5)	2.0 (2.0)	
***Pregnant at time of follow-up*** (3)	4 (1)	1 (1)	0 (0)	3 (1)	0 (0)	0.65
Mean gestation ± SD	8.7 ± 4.7	5.0 ± 0.0		10.5 ± 5.0		
***Currently taking antihypertensives*** (0)	106 (22)	49 (65)	8 (7)	26 (10)	23 (85)	<0.001 *
***Current daily smoker***, n (%)	8 (2) ^⋄^	3 (4)	2 (2) ^❖^	3 (1)	0 (0)	0.08 0.61
1–10 cigarettes per day, n (%)	6 (1)	3 (4)	1 (1)	2 (1)	N/A	
11–20 cigarettes per day, n (%)	1 (0)	0 (0)	0 (0)	1 (0)		
21 or more cigarettes per day, n (%)	0 (0)	0 (0)	0 (0)	0 (0)		
Don’t know, n (%)	0 (0)	0 (0)	0 (0)	0 (0)		
***Occasional smoker****,* n (%)	4 (1)	0 (0)	1 (1)	2 (1)	1 (4)	
***Past smoker****,* n (%)	107 (22)	21 (28)	30 (27)	53 (20)	3 (11)	
***Non-smoker***, n (%) ^⊕^	365 (75)	52 (68)	80 (71)	210 (78)	23 (85)	
***Don’t know****,* n (%)	0 (0)	0 (0)	0 (0)	0 (0)	0 (0)	
(0)						
***Current daily e-cigarette user****,* n (%)	2 (0)	2 (3)	0 (0)	0 (0)	0 (0)	0.19
***Occasional e-cigarette user****,* n (%)	4 (1)	1 (1)	0 (0)	3 (1)	0 (0)	
***Past e-cigarette user****,* n (%)	6 (1)	0 (0)	3 (3)	2 (1)	1 (4)	
***Non-e-cigarette user****,* n (%) ^⊕^	460 (95)	71 (93)	110 (97)	255 (96)	24 (89)	
***Don’t know****,* n (%)	10 (2)	1 (1)	0 (0)	7 (3)	2 (7)	
***Refused****,* n (%)	1 (0)	1 (1)	0 (0)	0 (0)	0 (0)	
(1)						
	** *Median (IQR)* ** ** * ^⋄^ * **					
***Number of children at time of 6 M follow-up*** (5)	1.0 (1.0)	2.0 (2.0)	1.0 (1.0)	1.0 (2.0)	2.0 (1.0)	<0.001 *

* Significant at *p* < 0.05 using ANOVA. ^⊚^ Those 25 women who selected more than one ethnicity were included as ‘other’ to preserve total sample sizes and accurate percentages. ^⋄^ Used for those outcomes which were not normally distributed. ^❖^ One of the women who reported to be smoking daily did not respond to further questioning about quantity smoked per day. ^⊕^ Includes those who have never used, or who have used a few times but never regularly. Abbreviations: 6 M, six month; CH, chronic hypertension; CH + PE, preeclampsia superimposed on chronic hypertension; E-cigarette, electronic cigarette; GH, gestational hypertension; HDP, hypertensive disorder of pregnancy; IQR, interquartile range; n, number; SD, standard deviation.

**Table 2 nutrients-15-03294-t002:** Lifestyle behaviours following HDP at 6 months postpartum as collected by the NSW Population Health Survey.

Lifestyle Behaviour (Missing)	Total (*n* = 484)	CH (*n* = 76)	GH (*n* = 113)	Preeclampsia (*n* = 268)	CH + PE (*n* = 27)	*p*-Value
**Nutrition**						
Vegetable serves/day, median [IQR] (0)	2.0 [2.0]	2.0 [2.0]	2.0 [1.8]	2.0 [2.0]	2.0 [1.5]	0.61
Meeting recommended 5 vegetable serves/day, n (%) (0)	44 (9)	5 (7)	11 (10)	25 (9)	3 (11)	0.85
Fruit serves/day, median [IQR] (0)	1.0 [1.0]	1.0 [1.5]	2.0 [1.0]	1.0 [1.0]	1.0 [1.0]	0.13
Meeting recommended 2 fruit serves/day, n (%) (0)	220 (46)	23 (30)	57 (50)	130 (49)	10 (37)	0.018 *
Meeting both recommended 5 vegetable serves and 2 fruit serves/day, n (%) (0)	31 (6)	2 (3)	9 (8)	18 (7)	2 (7)	0.51
Processed meat serves/week, median [IQR] (0)	1.0 [2.0]	1.0 [2.4]	1.0 [1.9]	1.0 [1.8]	1.0 [2.0]	0.49
Fried potato serves/week, median [IQR] (0)	1.0 [0.8]	1.0 [1.7]	1.0 [0.8]	1.0 [0.8]	0.7 [1.8]	0.43
Salty snack serves/week, median [IQR] (0)	0.5 [1.0]	0.5 [1.0]	0.5 [1.0]	0.5 [1.0]	1.0 [1.0]	0.63
Takeaway occurrences/week, median [IQR] (0)	0.5 [0.8]	1.0 [0.5]	0.7 [0.8]	0.5 [0.8]	0.5 [2.0]	0.25
Types of milk, n (%) (0)						0.06
Full fat	243 (50)	48 (63)	50 (44)	129 (48)	16 (59)	
Low/reduced fat	73 (15)	10 (13)	16 (14)	41 (15)	6 (22)	
Skim	48 (10)	5 (7)	12 (11)	30 (11)	1 (4)	
Evaporated or sweetened	0 (0)	0 (0)	0 (0)	0 (0)	0 (0)	
Almond/rice/oat milk	69 (14)	6 (8)	19 (17)	42 (16)	2 (7)	
Don’t drink milk	21 (4)	6 (8)	4 (4)	9 (3)	2 (7)	
Other	29 (6)	1 (1)	12 (11)	16 (6)	0 (0)	
Don’t know	1 (0)	0 (0)	0 (0)	1 (0)	0 (0)	
Cups of soft drinks/day, median [IQR] (0)	0.0 [0.4]	0.2 [1.0]	0.0 [0.4]	0.0 [0.3]	0.1 [0.6]	0.005 *
Cups of fruit juice/day, median [IQR] (0)	0.0 [0.3]	0.0 [0.3]	0.0 [0.3]	0.0 [0.3]	0.0 [0.3]	0.52
Cups of water/day, median [IQR] (0)	7.0 [4.0]	7.0 [6.0]	7.0 [4.0]	7.0 [3.0]	8.0 [4.0]	0.63
**Physical activity**						
Minutes spent doing vigorous household chores in the past week, median [IQR] (1)	0.0 [60.0]	5.0 [120.0]	0.0 [60.0]	0.0 [60.0]	0.0 [60.0]	0.004 *
Minutes spent doing vigorous yard work in the past week, median [IQR] (1)	0.0 [0.0]	0.0 [0.0]	0.0 [0.0]	0.0 [0.0]	0.0 [0.0]	0.27
Minutes spent continuously walking in the past week, median [IQR] (0)	180.0 [230.0]	120.0 [208.0]	190.0 [210.0]	180.0 [240.0]	140.0 [210.0]	0.19
Minutes spent doing vigorous physical activity in the past week, median [IQR] (0)	0.0 [60.0]	0.0 [37.5]	0.0 [97.5]	0.0 [60.0]	0.0 [10.0]	0.047 *
Including activity previously mentioned, minutes spent doing strength/toning activities in the past week, median [IQR] (3)	0.0 [45.0]	0.0 [45.0]	0.0 [60.0]	0.0 [40.0]	0.0 [0.0]	0.10
Combined minutes spent doing moderate–vigorous physical activity in the past week, median [IQR] (0)	210.0 [270.0]	150.0 [297.5]	240.0 [300.0]	235.0 [281.3]	180.0 [300.0]	0.11
Meeting recommended 150 min/week, n (%) (0)	313 (65)	40 (53)	81 (72)	178 (66)	14 (52)	0.023 *
Number of moderate–vigorous physical activity sessions in the past week, median [IQR] (0)	6.0 [6.0]	4.5 [6.8]	7.0 [6.0]	6.5 [6.0]	6.0 [4.0]	0.71
Meeting recommended 5 sessions/week, n (%) (0)	311 (64)	38 (50)	80 (71)	176 (66)	17 (63)	0.028 *
Meeting both recommended 150 min in 5 sessions/week, n (%) (0)	278 (57)	35 (46)	71 (63)	160 (60)	12 (44)	0.050 ^⦿^
Meeting recommendations for 150 min of moderate–vigorous physical activity in 5 sessions/week, as well as 5 vegetable and 2 fruit serves daily, n (%) (0)	22 (5)	1 (1)	6 (5)	13 (5)	2 (7)	0.469
**Alcohol**						
Alcohol intake occurrences/week, median [IQR] (0)	0.0 [1.0]	0.0 [0.3]	0.0 [1.4]	0.0 [1.0]	0.0 [1.0]	0.187
Standard drinks/week, median [IQR] (0)	0.0 [1.5]	0.0 [0.7]	0.0 [2.0]	0.0 [2.0]	0.0 [1.5]	0.592
Consumption of more than 4 standard drinks in one session in past 4 weeks, n (%) (207)	55 (20)	5 (14)	12 (17)	36 (23)	2 (20)	0.598

Conversions of data were made using the average value of 4.345 weeks per month, and 30.437 days per month. * Significant at level of *p* < 0.05 using ANOVA. ^⦿^ Borderline significant, *p* = 0.050 using ANOVA. Abbreviations: CH, chronic hypertension; CH + PE, preeclampsia superimposed on chronic hypertension; GH, gestational hypertension; HDP, hypertensive disorder of pregnancy; IQR, interquartile range; n, number.

**Table 3 nutrients-15-03294-t003:** Weight outcomes following HDP at 6 months postpartum.

Outcomes (Missing)	Total (n = 484)	CH (n = 76)	GH (n = 113)	Preeclampsia (n = 268)	CH + PE (n = 27)	*p*-Value
	** *Median [IQR]* **					
***BMI*** (kg/m^2^) (2)	26.6 [8.4]	30.2 [8.5]	26.8 [7.7]	25.5 [7.5]	27.3 [8.1]	<0.001 *
Overweight; BMI 25.0–29.9, n (%)	155 (32)	24 (32)	44 (39)	77 (29)	10 (37)	<0.001 *
Obese; BMI ≥ 30, n (%)	146 (30)	39 (51)	32 (28)	66 (25)	9 (33)	<0.001 *
***Waist-circumference*** (cm) (24)	90.0 [21.9]	101.5 [22.6]	90.5 [20.5]	87.0 [20.9]	94.0 [18.5]	<0.001 *
***Average systolic BP*** (mmHg) (6)	122.0 [15.0]	130.0 [16.0]	122.0 [12.0]	119.0 [14.5]	130.0 [21.0]	<0.001 *
Average systolic BP > 120, n (%)	280 (58)	65 (86)	68 (60)	126 (47)	21 (78)	<0.001 *
***Average diastolic BP*** (mmHg) (6)	81.0 [14.0]	88.0 [11.0]	81.0 [11.0]	79.0 [14.0]	88.0 [13.0]	<0.001 *
Average diastolic BP > 80, n (%)	266 (55)	63 (83)	62 (55)	120 (45)	21 (78)	<0.001 *
	** *Mean ± standard deviation* **					
***Body fat mass*** (%) (202)	37.0 ± 8.4	41.6 ± 9.2	37.7 ± 7.3	35.5 ± 8.4	35.6 ± 6.2	<0.001 *
	** *Number (%)* **					
***Returned to index pregnancy’s first antenatal appointment BMI or lower*** (7)	140 (29)	24 (33)	21 (19)	87 (33)	8 (30)	0.052

* Significant at level of *p* < 0.05 using ANOVA. Abbreviations: BMI, body mass index; BP, blood pressure; CH, chronic hypertension; CH + PE, preeclampsia superimposed on chronic hypertension; cm, centimetres; GH, gestational hypertension; IQR, interquartile range; kg, kilograms; mmHg, millimetres of mercury.

## Data Availability

The data presented in this study are available on reasonable request from the corresponding author. The data are not publicly available due to ethics approval conditions.

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
