# Peer review of "The Association between Nutrition, Physical Activity, and Cardiometabolic Health at 6 Months following a Hypertensive Pregnancy: A BP2 Sub-Study"

_nutrients, 2023, doi:10.3390/nu15153294_

Round 1

Reviewer 1 Report

Comments for the authors:
In this article, the authors investigated the association between diet, physical activity and cardiometabolic health 6 months following a hypertensive pregnancy: a BP2 sub-study.

This study is well-designed, but the results are not surprising, so the novelty of the article is not significant. An unhealthy lifestyle is a serious public health problem, as it significantly increases the risk of obesity and chronic diseases such as cardiovascular disease, type 2 diabetes, hypertension and certain cancers. It is not surprising that mothers with high blood pressure or pre-eclampsia do not follow a healthy lifestyle after pregnancy. Unfortunately, this is a common social problem.

Minor Remarks:
-    What could be the practical significance of their results?
-    Please make suggestions in the discussion section on how to improve mothers' attitudes toward physical activity and healthy lifestyles!
-    The healthy control group is missing in this study. Do you have information on whether pregnant mothers without complications change their lifestyle after pregnancy? Do they exercise more or eat healthier?

Reviewer 2 Report

GENERAL COMMENTS

The study aims at describing lifestyle behaviors in women 6-months post-HDP. However, it takes into consideration only nutrition and physical activity. It is not clear if tobacco and alcohol use is taken into consideration, despite their association with CVDs hence part of healthy lifestyle interventions. Furthermore, antenatal care (in terms of content, number, and timing of antenatal care visits and diagnostics) and safe motherhood (including education and counseling sessions during pregnancy) are not taken into consideration either. It is not clear if these were part of the study across all materials and methods (they are mentioned only in reference to the Population Health Survey), and they are not consistently captured throughout the study’s introduction, results, and discussion. For example, alcohol is not captured in Table 2 “study participant demographics”. Please clarify and, if needed, reflect potential gaps in the limitations (section to be added).

Findings in terms of correlation with lifestyle behaviors and HPDs are mostly not surprising (except alcohol and smoking as per tables 1 and 2) and the added value of this study could be to leverage them to inform and guide interventions based on education and counseling (i.e. the three intervention arms), but there is no mention about their implementation and outcomes.

The study employes a number of materials and methods but they are not clearly identified and described. Please confirm the following list and ensure adequate description (development, implementation, and analysis) of each item: RCT with three study arms, medical records, a pre-randomisation questionnaire (is it the 6M BP2 questionnaire?), and a population health survey over the phone. Please add and describe any other item.       

The reference period should be stated in the main text, not only in Figure 1. Please provide information in the main text about: When was the study conducted? When were participants enrolled? What is the start and end dates of records included in the database? Please provide clear information about the study’s timeline, both overall and for each respective segment/item.

Correlation was calculated between adherence to recommendations for diet/physical activity and cardiometabolic outcomes, but not with determinants, for instance: HPD diagnosis by (specific) gestation duration, and BMI classification. Why so?  

No correlation was identified between HDP and smoking, nor with alcohol intake (tables 1 and 2). These findings are not in line with the evidence base, and it would be good to provide potential explanation(s).

The study does not have a Limitations section, please provide it.

Throughout the study, it would be helpful to specify if statements are at global level and/or the Australian setting specifically.

References need some improvement.

SPECIFIC COMMENTS

Introduction

·       Lines 36-39: This sentence presents a number of important statistics about HDP and pregnancy complications, with a total of 7 references. As currently presented, locating the specific statistics in the corresponding reference is not user-friendly. Please consider providing specific references for each statistics’ statement.

·       Lines 40-41 and 53-54: The three statements about the global burden of maternal and perinatal deaths as well as pre-term and small-for-gestational age births due to preeclampsia, and about CVD as a leading cause of mortality in women would benefit from an additional reference from a global institutional source (such as WHO, FIGO, or other relevant).

·       lines 48-19: Besides reference #6, it would be helpful to add one or two additional references about CVD risk after adjusting for co-founding factors.

·        Lines 58-59: The study’s relevance would benefit from adding reference to relevant recommendation(s) issued by WHO.

·       Lines 62-63: Please specify if this relates to standard postpartum management at global level, or only Australian level.

·       Lines 63-67: This sentence about “Diets such as Dietary Approaches to Stop Hypertension and exercise recommendations (150mins per week of moderate-to-vigorous activity in five sessions” is accompanied by total 4 references (#2,5,14,18). However, only reference #18 partially (“150mins per week …” but not “the Dietary Approaches…”). Please check and correct if/as needed.

·       Lines 71-72: Besides reference #5, it would be helpful to add one or two additional references about micronutrient issues and CVD.

Materials and Methods

The section does not provide sufficient information about questionnaire development, administration, and response tabulation. Please provide. In particular, please ensure that information is provided about how the open-ended fields in the questionnaire were handled, especially in terms of cleaning, standardization, and tabulation before statistical analysis.

Participant eligibility and recruitment

Lines 116-117: Please provide information about participation of women from non-English speaking background through interpreter services and translated study materials. For instance, what was the proportion of study participants that were non-English speakers?

Statistical analysis

Line 159: please explain briefly what is the “central BP2 database”.

Impact of COVID-19

Missing data is mentioned but not specified. Please provide information about their number, type, and timing of missing data. Furthermore, please specify what proportion of missing data is compatible with the COVID-19 pandemic hence potentially explained by it. This is important as study recruitment and consent occurred between January 1st 2019 and August 9th 2022, hence only partially compatible with the COVID-19 pandemic.

Missing data should be reflected in the Limitations section, to be added.

Results

Line 184: how many women were eligible for enrolment in the study? This should be stated, before the number of women who consented to participate in the study.

Line 190: “final numbers” is not adequate wording, please revise.

Lines 197-200: please quantify “majority”, “over two thirds”, “significantly younger”, “higher rates of first pregnancy” (either in numbers in parentheses or in the main text).

Lines 198-199: what is the meaning of “women following preeclampsia”?

Lines 201-204: please qualify “differences were seen between highest level of education, gestation at diagnosis and birth, booking BMI, maternal days spent in hospital at birth and number of children at time of follow-up”.

Line 202-203: what is “booking BMI”?

Line 204: what is the meaning of “At booking for the index pregnancy”?

Line 207: “spent longer in hospital during 207 their birth admission” can be reworded.

Table 1:

-          “Demographic (missing)”: what does it mean?

-          Ethnicity (0), Highest education (4), Index pregnancy first baby (0), etc…: what is the meaning of values in parentheses?

-          Line 219: assigning to “other” those who selected more than one ethnicity loses information about ethnicity, even if preserves total sample sizes and accurate percentages. Would it be possible to provide this information, perhaps in footnote?

Table 2:

“Lifestyle behaviour (missing)”: what does it mean?

Discussion

Lines 375-380: pity the questionnaires did not collect information about drivers and barriers of physical activity and healthy nutrition, as such information would have been instrumental to inform and guide the 3 interventions arms. This gap may be reflected in the Limitations?

Critique of experimental design and future directions

Lines 421-483: please specify in what way “the present study is the first of its kind to describe lifestyle behaviours, cardiometabolic profiles and the correlations between them for women soon after HDP”.

Lines 484-486: please specify to what extend the study findings may be generalizable: to Sydney? To Australia?

6M BP2 Questionnaire

-          Education: the question on participant’s education does not capture levels below secondary school. Furthermore, the question on partner’s education includes an additional option (“not applicable”), which is not similarly available for the participant.

-          Many questions are asked in open-ended fields, rather than dichotomized options which would be better for analysis. For instance: occupation, current medication, illness history, fertility issues, partner change, HBP details, GD details, end of breastfeeding, reason(s) for pride in labor/birth, reason(s) for unhappiness in labor/birth, post-partum BP visit details, post-partum health problems visit details, referral details, mental health details, new medications details. As explained in comments under the Materials and Methods section, please provide information about how the open-ended fields in the questionnaire were handled, especially in terms of cleaning, standardization, and tabulation before statistical analysis.

-          As mentioned in general comments, Tobacco and alcohol use is not taken into consideration, nor is access to/coverage of antenatal care (in terms of content, number, and timing of antenatal care visits and diagnostics), nor is access to/participation in safe motherhood groups including education and counseling during pregnancy.

Population health survey Questionnaire

Where in the manuscript is data collected through this questionnaire presented?

some wording is unclear and/or inadequate, please see specific comments for revisions.

Reviewer 3 Report

Very informative and content text, nicely processed, many intersting data, but disscussion is too long and extensive and it would have to be summerized.

There is an error in Table 3 row 6. where it say "Average sistolic BP>80, and it should be Average diastolic BP>80...
